# Geographic name resolution service: A tool for the standardization and indexing of world political division names, with applications to species distribution modeling

Bradley L. Boyle[1]*, Brian S. Maitner[2], George G. C. Barbosa[1], Rohith K. Sajja[1], Xiao Feng[3], Cory Merow[2], Erica A. Newman[4], Daniel S. Park[5,6], Patrick R. Roehrdanz[7], Brian J. Enquist[1,8]

1 Department of Ecology and Evolutionary Biology, University of Arizona, Tucson, AZ, United States of America, 2 Eversource Energy Center and Department of Ecology and Evolutionary Biology, University of Connecticut, Storrs, CT, United States of America, 3 Department of Geography, Florida State University, Tallahassee, FL, United States of America, 4 School of Natural Resources & the Environment, University of Arizona, Tucson, AZ, United States of America, 5 Department of Biological Sciences, Purdue University, West Lafayette, IN, United States of America, 6 Purdue Center for Plant Biology, Purdue University, West Lafayette, IN, United States of America, 7 The Moore Center for Science, Conservation International, Arlington, VA, United States of America, 8 The Santa Fe Institute, USA, Santa Fe, NM, United States of America

* bboyle@arizona.edu

**Data Availability Statement:** A publicly available instance of the GNRS API can be accessed directly

## Abstract

Massive biological databases of species occurrences, or georeferenced locations where a species has been observed, are essential inputs for modeling present and future species distributions. Location accuracy is often assessed by determining whether the observation geocoordinates fall within the boundaries of the declared political divisions. This otherwise simple validation is complicated by the difficulty of matching political division names to the correct geospatial object. Spelling errors, abbreviations, alternative codes, and synonyms in multiple languages present daunting name disambiguation challenges. The inability to resolve political division names reduces usable data, and analysis of erroneous observations can lead to flawed results. Here, we present the Geographic Name Resolution Service (GNRS), an application for correcting, standardizing, and indexing world political division names. The GNRS resolves political division names against a reference database that combines names and codes from GeoNames with geospatial object identifiers from the Global Administrative Areas Database (GADM). In a trial resolution of political division names extracted from >270 million species occurrences, only 1.9%, representing just 6% of occurrences, matched exactly to GADM political divisions in their original form. The GNRS was able to resolve, completely or in part, 92% of the remaining 378,568 political division names, or 86% of the full biodiversity occurrence dataset. In assessing geocoordinate accuracy for >239 million species occurrences, resolution of political divisions by the GNRS enabled the detection of an order of magnitude more errors and an order of magnitude more error-free occurrences. By providing a novel solution to a significant data quality impediment, the GNRS liberates a tremendous amount of biodiversity data for quantitative biodiversity

at https://gnrsapi.xyz/gnrs_api.php, or indirectly using the GNRS R package or GNRS web interface. The GNRS R package can be downloaded from GitHub (https://github.com/EnquistLab/RGNRS) using the devtools package [60], from CRAN (see https://cran.r-project.org/web/packages/GNRS/index.html). The GNRS web interface can be accessed at https://gnrs.biendata.org. Source code for the GNRS database, core services and API are available from the GNRS GitHub repository (https://github.com/ojalaquellueva/gnrs). Example scripts demonstrating how to call the API in R and PHP are available from the API subdirectory of the GNRS repository (https://github.com/ojalaquellueva/gnrs/tree/master/api). Code to import GADM content to PostgreSQL is available from https://github.com/ojalaquellueva/gadm. Code to import GeoNames is available from https://github.com/ojalaquellueva/geonames. Source code for the GNRS web interface is available from https://github.com/EnquistLab/GNRSweb. All source code for all GNRS components is freely available under MIT licenses. Access to the complete GNRS database is governed by the licenses of the contributing databases. Limitations imposed prohibit us from directly redistributing the complete copies of the GNRS database. However, users can build an identical copy of the GNRS database using the GNRS source code with data obtained directly from the contributing databases. Step-by-step instructions are provided in the GNRS GitHub repository. An example data file for testing the GNRS is available here: https://github.com/ojalaquellueva/gnrs/blob/master/data/user/gnrs_testfile.csv. Data and code for replicating the GNRS case study (Fig 3 and all summary statistics) are available here: https://doi.org/10.5281/zenodo.6370837.

**Funding:** This work was supported by US National Science grant HDR-1934790. The funders had no role in study design, data collection and analysis, decision to publish, or preparation of the manuscript.

**Competing interests:** The authors have declared that no competing interests exist.

research. The GNRS runs as a web service and is accessible via an API, an R package, and a web-based graphical user interface. Its modular architecture is easily integrated into existing data validation workflows.

## Introduction

Large databases of georeferenced species occurrences (GSOs) fuel an increasingly diverse body of research into past, current, and future patterns of species distributions and traits [1]. GSOs provide essential inputs for species distribution models (SDMs) [2–5], which in turn have been used to predict relative vulnerability of species and populations to climate change [6], identify priority conservation strategies [7] and assess the biodiversity impacts of policies governing land use, deforestation and burning [8]. SDMs and the GSOs from which they are derived are helping to clarify distributions of disease vector organisms and identify new disease hotspots [9, 10]. GSOs and associated trait data from museum specimens have been used to disentangle patterns of temporal and spatial change in body size of birds [11] and melanism in butterflies [12]. Given the breadth of applications of SDMs, they must be robust, which in turn depends on the accuracy of the species occurrence data that inform them. The challenge is identifying, differentiating, and correcting erroneous or inaccurate geographic distribution information.

The fitness of GSOs for such analyses hinges on the accuracy of the associated location data. Despite recent advances in automated tools for standardization and correction of errors, the potential presence of erroneous or inaccurate geo coordinates in biodiversity "big data" remains a major concern [13, 14]. A widely used method for assessing reliability of coordinates is to check if they fall within the boundaries of their associated political divisions (hereafter, "political geovalidation"). A point falling outside a declared political division is flagged for inspection and either corrected or rejected [15]. Another common validation links a GSO via its declared political division to one or more country, state or county taxonomic checklists to determine if the species is native or introduced in the region of observation; observations of introduced species may be excluded from further analysis [16], unless modeling of invasive species distributions is the focus of the research [17]. A surprising impediment to these otherwise simple validations is lack of standardization among political division names, identifiers, and hierarchies.

The importance of political divisions as both units of data aggregation and data quality pitfalls extends well beyond GSOs and SDMs. Analysis of relationships between environmental factors, health care policy, and health care outcomes are a mainstay of public health research, with many studies relying on data aggregated by first- and second-level administrative divisions [18]. Multi-country comparisons of crime statistics aggregated at subnational levels are common in criminology and sociological research [19]. A recent study of human reliance on protected natural areas throughout the global tropics combined geospatial information on protected areas with household survey data aggregated by subnational administrative units [20]. Incomplete or inconsistent standardization of political division names and identifiers increases the burden of data aggregation, especially when historical data are involved [21, 22].

A promising way forward is the development of a general tool for the standardization of political division names, identifiers, and hierarchies. However, this goal is complicated by the myriad of alternative names, spellings and abbreviations used to refer to the same country or subnational unit. In addition, geographical data processing codes such as ISO (International

Organization for Standardization; [23], FIPS (the United States' Federal Information Processing Standards; [24], and HASC (Hierarchical Administrative Subdivision Codes; [25]) may be used instead of names. Multiple languages, accented characters, and different character set encodings provide additional layers of complexity. Spelling errors may also be introduced during data entry. Together, these issues represent a daunting disambiguation challenge on par with taxonomic name resolution [26, 27]. Failure to resolve political division names can lead to data loss by excluding GSOs of unknown quality or the inability to georeference historical observations [28]. Naive use of unvalidated GSOs can result in misleading, erroneous, or biased research results [29].

Resolution of political division names and identifiers is a subset of the broader topic of toponym resolution—the disambiguation of geographic place names [30]. The recognition and resolution of toponyms in natural language text has long been a focus of natural language processing (NLP) research [31]. A variety of toponym resolution algorithms and services for natural language text are currently available [31, 32], with recently-developed machine learning approaches involving recurrent neural networks achieving high accuracy rates [33, 34]. However, there are several reasons why existing NLP solutions may be suboptimal for the narrower use case of political division name resolution of GSOs. First, input data derived from GSOs are not natural language text but rather structured data, typically downloaded from biodiversity observation databases such as GBIF [35]. The identity of the data elements as administrative divisions is known in advance, along with their positions in the administrative division hierarchy. Thus, the entity recognition step performed by many services is unnecessary and may inject uncertainty where none exists. Second, the hierarchical relationships of a set of two or more nested political divisions derived from a GSOs necessarily constrain and inform resolution at lower levels of the hierarchy. Although heuristics involving hierarchy have been shown to degrade accuracy of toponym resolution of natural language text and are not universally employed by NLP toponym resolution algorithms [31], hierarchical searching is essential for disambiguating structured political division names associated with GSOs, in much the same way as it is fundamental to taxonomic name resolution [27, 36]. Third, the ultimate target of GSO political division name resolution is generally not the name itself, but rather the spatial object needed to verify that the GSO's coordinates fall within the declared political division. Although existing services such as GeoTxt [31] and the GeoNames' API [37] resolve political division names against GeoNames, the additional step of discovering the equivalent spatial object identifiers in databases such as GADM is left to the user—a significant challenge in itself, as we explain below (see **Reference database**). A recent effort to add GeoNames and GADM identifiers to administrative divisions within Wikidata [38, 39] should eventually bridge the gap between these critically important resources. At the time of writing, however, indexing is still incomplete, with many political divisions in Wikidata lacking both GeoNames and GADM identifiers (e.g., [40]).

Here, we describe a software tool for correcting, standardizing, and indexing world political division names, the Geographic Name Resolution Service (GNRS). The GNRS accepts one or more 1–3 level political division name combinations (country, country+state or country+state +county, or equivalent), and resolves them against GeoNames [41] and the Database of Global Administrative Areas (GADM; [42]), supplemented with additional names and codes from Natural Earth [43]. For each name resolved, the application returns the standard GADM name, a plain-ascii English-language name minus class identifiers (e.g., "State of", "Provincia de", "Département"), ISO codes, and Geonames and GADM identifiers. Match scores and summaries describing how the submitted name was matched and overall matching completeness are also returned with the resolved name. Other GNRS options support retrieval of alternative names in multiple languages and character sets from Geonames and GADM.

**Table 1. BIEN data validation and standardization tools.**

| Service | Purpose | Interfaces |
|---|---|---|
| Taxonomic Name Resolution Service (TNRS) [27] | Resolve taxonomic names against one or more authoritative taxonomic sources. | Shell, API, R package, web interface [46] |
| Geographic Name Resolution Service (GNRS) | Resolve political division names to GeoNames and return GADM spatial object identifier. | Shell, API, R package, web interface [47] |
| Geocoordinate Validation Service (GVS) | Flag coordinates invalid or in ocean, Return containing political divisions and flag likely centroids. If coordinates accompanied by declared political division, will resolve using GNRS and flag geocoordinates falling outside one or more political divisions. | Shell. Public interfaces under development |
| Native Species Resolver (NSR) | Determine if taxon is native or introduced in the political division of observation. Requires prior name resolution by TNRS. | Shell. Public interfaces under development |
| Cultivated Plant Detection Service (CPDS) | Searches specimen descriptions and analyzes locality information to determine if observation is likely a cultivated plant (human planted or maintained). | Shell. Public interfaces under development |

Standardized political division names and GADM identifiers can be used to retrieve spatial objects from GADM to perform political geovalidation of GSOs, or to submit the GSO to other validation services such as the BIEN Native Species Resolver ([16]; Table 1).

Despite the availability of open access databases of administrative division spatial objects such as GADM and global gazetteers of names such as GeoNames, resolving political division names to their corresponding spatial representations remains challenging due to the lack of standardization of object names and their identifiers, and incomplete linkages among reference data sources. To our knowledge, no existing service links these sources comprehensively and provides informatics tools for resolving large volumes of unstandardized data against them. Our goal in developing the GNRS is to fill this gap.

## Overview of the GNRS

### Architecture

Originally developed as part of the BIEN database pipeline [16, 27, 44, 45], the GNRS is one of a series of data validation and standardization tools that we are making available to the biodiversity research community as modular web services (Table 1). Each service will be accessible through various interfaces, using standardized plain text input and output that allows multiple services to be chained together into more complex validations. We are releasing these services as standalone applications to enable independent development of each service and to encourage scrutiny and improvement of algorithms and data by the community.

All BIEN validations services share the architecture shown in Fig 1. Components of the architecture include: (1) a core service in which user data are standardized against a reference database using algorithms implemented primarily in SQL; (2) a data integration application that builds and periodically updates the reference database (a partly normalized "data warehouse" *sensu Inmon* [48]) from external sources, (3) a controller layer which manages concurrent requests and implements parallelization using Makeflow [49]; (4) an administrative interface which interacts directly with the controller; (5) a JSON-based application programming interface (API) which supports large input-output data payloads; (6) an R package; and (7) a web-based graphical user interface. The API handles all public access to the core service, including via the R package and web interface. The GNRS runs in the Linux environment and was developed under Ubuntu 16.04.7 LTS [50].

Several design elements of the BIEN validation service architecture optimize the processing of large data sets within a multi-user environment. These include parallel processing and caching of previous results. For more details of performance features see S1 Appendix.

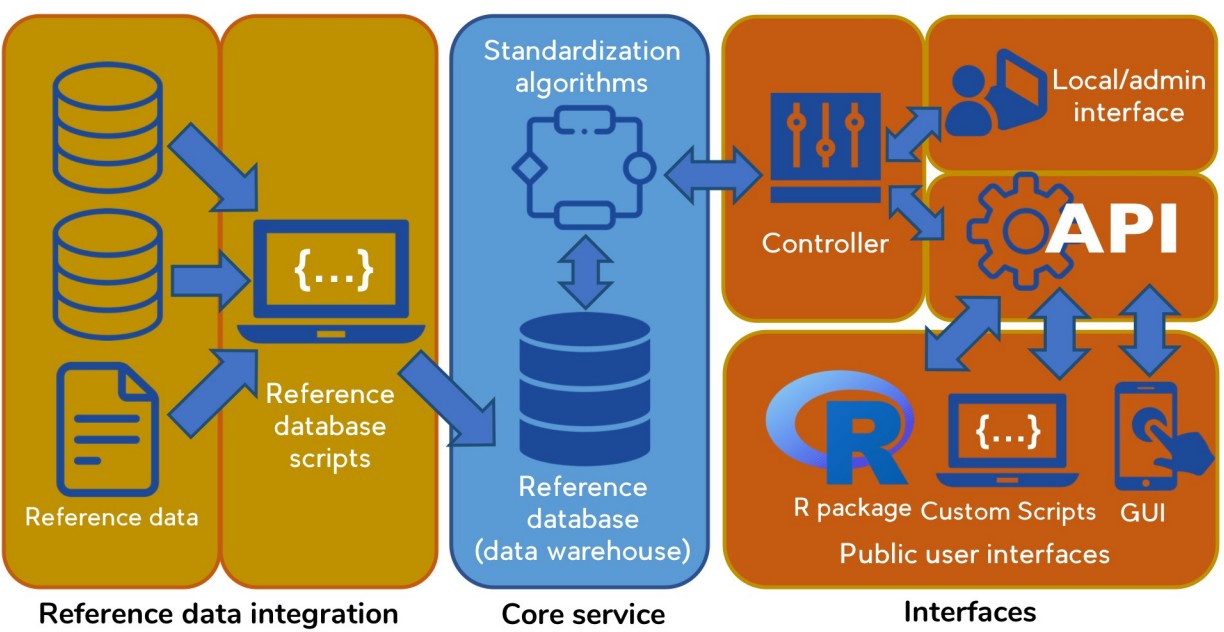

**Fig 1. BIEN validation service architecture, as implemented for the GNRS.** Reference data are stored locally within the core service as a periodically-updated, versioned data warehouse. A controller manages parallelization and optimization of concurrent requests. Interfaces include a JSON API, an R package, and a web-based graphical user interface. The API handles all public interaction with the core service.

Users from the biodiversity informatics community will see many parallels between the BIEN suite of data validation services and the Global Names Architecture (GNA) [51]. However, despite the considerable overlap in services such as taxonomic name resolution, the goals and domains of the two projects differ in important ways. GNA is broadly concerned with taxonomy and offers a range of tools for the indexing of taxonomic names and taxonomic literature [26], the discovery and resolution of taxonomic names in natural language text (particularly in scientific literature), and to a lesser extent, the exchange and management of biological observation data via standards such as Darwin Core [52]. Whereas the domain of GNA encompasses the entire tree of life [26], BIEN is more narrowly focused on plants. The services it provides address different aspects of the validation and standardization of structured biodiversity observation data [16]. More specifically, the GNRS and the related Geocoordinate Validation Service (GVS; see Table 1) are tools for resolving errors in locality data and coordinates associated with georeferenced biological observations and are thus more akin to data cleaning applications such as [15] and [53]. In our view, overlap with existing applications should be seen not as a redundancy but as an opportunity for scrutiny and improvement of approaches to cleaning and standardizing biodiversity data.

## Reference database

Political division names are resolved by the GNRS against a reference database consisting of the names, codes and identifiers of all countries plus their first-level (state/province) and second-level (county/parish) administrative divisions in GADM (for a discussion of why the GNRS currently resolves to 3 levels of political division hierarchy, see **Caveats**). To enable resolution of a broader range of alternative names, formulations and spellings, each GADM political division is linked to a lookup table of official and alternative names, codes and abbreviations in multiple languages from Geonames, supplemented with additional codes

from the Natural Earth and a custom list of common name variants prepared by the authors (the latter included in the GNRS source code repository; see https://github.com/ojalaquellueva/gnrs/tree/master/gnrs_db/data). Names, codes, and identifiers for country-, state- and county-level political divisions from these sources are merged within a single PostgreSQL database [54] by a pipeline of SQL statements managed by Bash shell commands [55]. Because GADM and GeoNames lack shared administrative division codes, Natural Earth was used as a crosswalk to align the two databases, using geoname_id to link to GeoNames and HASC codes to link to GADM. Unmatched and missing codes at lower political division levels resulted in large numbers of administrative divisions requiring *ad hoc* scripting or manual inspection to link the two databases fully. The steps and challenges involved in merging these data sources are described in S2 Appendix.

We appreciate that many territories are subject to conflicting claims by different jurisdictions, and that the configuration of national and subnational units returned by the GNRS may conflict with public sentiment, official policy, or both in some countries. The GNRS has no opinion on these matters but simply transmits the configuration of countries and their subdivisions represented in GeoNames, GADM, and Natural Earth. As GNRS code is fully open source, we encourage users dissatisfied with aspects of the data currently served to develop new instances of the GNRS, using alternative data sources as needed.

## Metadata

Management of metadata within the GNRS database and transmission via user interfaces follows the principles established by the BIEN database and its public interface, the BIEN R package [16]. Summary tables within the GNRS reference database manage information on reference data sources and the GNRS itself. Versions, access date, source URLs, project websites, and bibtex-formatted citations for GADM, GeoNames, and NaturalEarth are stored in table "source". Metadata on the GNRS (database release date, code version and citation) is maintained in table "meta". Metadata on other contributors of resources or data is stored in table "collaborator". All metadata can be queried via the API using routes "source", "meta" and "collaborator"; this information is also exposed by the GNRS R package and the GNRS website.

## User input

The basic input for the GNRS is a 1- to 3-level political division combination (PDC) consisting of a country, a 1st-level administrative division (state, province, department, etc.) and a 2nd-level administrative division (county, parish, municipality, etc.), separated by commas. The country is required but 1st- and 2nd-level divisions are optional; however, a 1st-level division must be present if a 2nd-level division is supplied. Both comma delimiters must be present, even if one or more administrative division is absent. Names that contain commas must be surrounded by double quotes. Each PDC must be on its own line. Examples of data suitable for input to the GNRS are shown in Table 2.

One or more PDCs in this format can be submitted directly to the GNRS web interface by pasting them into the input box (Fig 2). The input format for the GNRS API and R package is similar to the basic format, except for an additional, user-supplied unique identifier ("user_id") in the first column (i.e., user_id, country, state_province, county_parish). A single-column identifier provides a reliable way of joining the multi-column GNRS output back to databases, where NULL values in some fields may result in failed joins or data loss. Use of identifiers is optional; however, the four column format must be maintained in the order described, with all three comma delimiters present on all lines, as shown in the API examples

**Table 2. Input format for political divisions submitted to the GNRS via the web user interface and API.**

| Interface | Examples |
|---|---|
| Web | USA,Arizona,Pima County |
| | México,Oaxaca, |
| | Costa Rica, |
| | Guyana,Upper Takutu-Upper Essequibo,"Yakarinta—Wowetta, Surama" |
| API (with id) | 1,USA,Arizona,Pima County |
| | 2,México,Oaxaca, |
| | 3,Costa Rica, |
| | 4,Guyana,Upper Takutu-Upper Essequibo,"Yakarinta—Wowetta, Surama" |
| API (no id) | ,USA,Arizona,Pima County |
| | ,México,Oaxaca, |
| | ,Costa Rica, |
| | ,Guyana,Upper Takutu-Upper Essequibo,"Yakarinta—Wowetta, Surama" |

Format requirements for the GNRS R package are the same as the API (see documentation).

in Table 2. Data are submitted to the API as JSON attached to the body of a POST request. The R package automatically handles the conversion to JSON and construction of the API request.

## Name resolution workflow

PDCs are resolved by working down the 3-level political division hierarchy beginning with the country. The algorithm first tries matching by code (ISO, FIPS, HASC) before attempting to

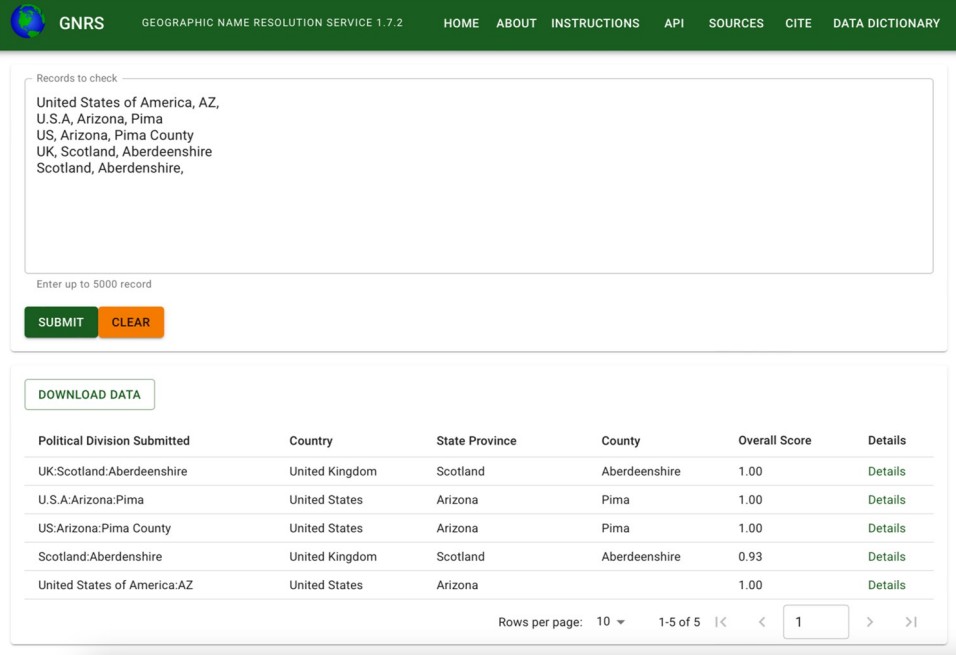

**Fig 2. Screenshot of the GNRS web user interface.** Comma delimited political divisions are pasted into the top input box. Paginated results, displayed below the input, can be sorted by any column and downloaded as comma- or tab-delimited files. The "Details" hyperlink at the end of each row displays the full results for that row, along with field definitions from the GNRS data dictionary.

match unresolved political divisions by standard and alternative names. After all exact matching methods have been exhausted, fuzzy matching of the remaining unresolved names is attempted using the Postgres implementation of trigram similarity [56]. The GNRS uses a default trigram match threshold of 0.5, a conservative setting that favors avoiding false positives. The match threshold can also be adjusted on the fly by the users of the R package or API. At each step, the match method used for a successful match is saved and returned to the user in fields "match_method_country", "match_method_state_province" and "match_method_-county_parish" (example values: "iso code", "exact match standard name", "fuzzy match alternate name"; see Tables 2–4 in S6 Appendix for more examples). A match score from 0 to 1 (where 0 ="no match" and 1 ="exact match") is calculated as the trigram similarity between the name submitted and the name matched and saved to fields "match_score_country", "match_score_state_province" and "match_score_county_parish". After all matching steps have been completed, an "overall_score" for the PDC as a whole is calculated as the average match score of all component political divisions. One of three descriptors of overall match completeness ("no match", "partial match", "full match") is saved to field "match_status". The GNRS also returns the political division level submitted and the political division level matched, using terms "country", "state_province" and "county_parish".

Two classes of political divisions not resolved by the default GNRS workflow required custom solutions. These are (1) territories and other subnational geopolitical units treated as countries ("states-as-countries") by the reference databases, and (2) countries belonging to multinational unions, with the latter treated as countries and its member countries treated as first-level divisions ("countries-as-states"). An example of state-as-country is Puerto Rico, an unincorporated territory of the United States which is treated by GADM and GeoNames as a top-level political entity with ISO code PR, but frequently recorded in biodiversity data as a state-level division of the United States (e.g., "USA, Puerto Rico"). Examples of countries-as-states are England, Scotland and other member countries of the United Kingdom, which are treated as 1st-level political divisions of the UK by GADM and GeoNames, but also appear as countries in biodiversity data. The GNRS solutions to these special cases are described in S4 Appendix.

## Interfaces

**Linux command line.** Developers installing their own instance of the GNRS can invoke the GNRS directly from the Linux shell using commands gnrs_batch.sh (single batch mode) or gnrs_par.pl (parallel mode). See the main README in the GNRS GitHub repository for documentation of syntax and usage examples (https://github.com/ojalaquellueva/gnrs/blob/master/README.md). Accessing the GNRS directly via the shell bypasses the API and its default limit of 5000 rows per request, thus enabling the processing of enormous data sets in a single operation.

**GNRS API.** All public interaction with the GNRS—including requests from the GNRS R package and GNRS website—is handled by a JSON-based API (see [57] with a single route (https://gnrs.biendata.org/gnrs_api.php). Different endpoints and parameters (Table 3) are specified in JSON object "opts" (options). Request data are converted to JSON object "data". Objects opts and data are combined into a single nested JSON object and attached to the body of the POST request submitted to the API.

API endpoint "resolve" performs name resolution of the political divisions contained in the POST data; it supports a single optional parameter "tfuzzy" which accepts a numeric value between 0 and 1 and allows the user to vary the default trigram fuzzy match score threshold. Other API endpoints include a data dictionary defining all name resolution output fields; detailed lists of names, alternate names, codes, and identifiers of countries, states, and counties;

**Table 3. GNRS API endpoints and their meanings.**

| Endpoint | Purpose | Data |
|---|---|---|
| resolve | Resolve submitted political division and return standardized names and identifiers | One or more sets of political divisions (country plus up to 2 lower political divisions) optionally preceded by user-supplied record identifiers |
| countrylist | Return names and identifiers of all countries in the GNRS database | None |
| statelist | Return names and identifiers of all states in submitted countries. | List of countries (name) for which to return states. Get country names from route "countrylist" |
| countylist | Return names and identifiers of all counties in submitted states. | List of states (GNRS identifier state_province_id) for which to return counties. Values of state_province_id from route "statelist" |
| meta | Return metadata on the GNRS | None |
| sources | Return metadata on reference data sources used by the GNRS | None |
| citations | Return citations for the GNRS and all data sources | None |
| dd | Return definitions of output fields (data dictionary) | None |

Endpoints, parameters and input data are attached to the body of the POST request as a nested JSON object, with the endpoint and its parameters in element "option" and input data in element "data".

and metadata and citations for the GNRS and its sources. Descriptions of all API endpoints are provided in '. Example scripts demonstrating calls to the GNRS API in R and PHP are provided in the api subdirectory of the GitHub GNRS repository (https://github.com/ojalaquellueva/gnrs/tree/master/api). Results are returned to the user as JSON. The GNRS API is written in PHP [58].

**GNRS R package.** The R package GNRS provides a family of functions for interacting with the GNRS API using the R language [59]. All major functionality available by calling the GNRS API directly is available through the R package (Table 4). GNRS can be installed from CRAN using the command install.packages("GNRS") or the development version can be

**Table 4. GNRS R package functionality.**

| API Option | R function | Input data | Purpose |
|---|---|---|---|
| resolve | GNRS() | Political division dataframe containing 4 columns: user_id, country, state_province, and county_parish. Number of batches (Optional) | Resolve submitted political division and return standardized names and identifiers |
| resolve | GNRS_super_simple() | country, state_province (Optional), county_parish (Optional), user_id (Optional) | Resolve submitted political division and return standardized names and identifiers |
| countrylist | GNRS_get_countries() | None | Return names and identifiers of all countries in GNRS |
| statelist | GNRS_get_states() | country_id (Optional) | Return names and identifiers of all states, or states in submitted countries (if country_id is supplied). |
| countylist | GNRS_get_counties() | state_province_id (Optional) | Return names and identifiers of all counties, or counties in submitted states (if state_province_id is supplied). |
| meta | GNRS_version() | None | Return version metadata on the GNRS |
| sources | GNRS_sources() | None | Return metadata on reference data sources used by the GNRS |
| citations | GNRS_citations() | None | Return citations for the GNRS and all data sources |
| dd | GNRS_data_dictionary() | None | Return definitions of output fields (data dictionary) |
| meta, sources, citations | GNRS_metadata() | bibtex_file (Optional) | Wrapper function that returns metadata on version, sources, acknowledgments, and citations. |
| | GNRS_template() | nrow (Optional) | Returns an empty dataframe of nrow (default is 1) rows that can be populated by users. |

installed directly from the GNRS GitHub repository (https://github.com/EnquistLab/RGNRS) using the devtools package [60] with the command devtools::install_github("EnquistLab/ RGNRS"). The GNRS package relies on the package httr [61] to interact with the API, jsonlite [62] to convert to and from json, and the packages knitr, rmarkdown, devtools, and testthat [60, 63–65] for development and testing. GNRS R package functions begin with the prefix "GNRS_. . ." to simplify function location through tab-completion.

**GNRS web interface.** The GNRS website is a graphical user interface to the GNRS that runs on both desktop and mobile devices (Fig 2). Political divisions are pasted or typed directly into an input box and results are displayed below. Results may be downloaded in comma-delimited (CSV) or tab-delimited (TSV) formats. With the exception of user IDs, which are not supported, most functionality available via the GNRS API and GNRS R package is also available through the GNRS website. Metadata served via API options "meta", "sources", "citations" and "dd" (data dictionary) are displayed on pages "Cite", "Sources" and "Data dictionary". The GNRS website was developed using the open-source Next.js framework [66], written in JavaScript with the back-end using runtime and the front-end using the React library [67]. The interface was designed using Material-UI, an open-source React-component library that follows Material Design principles [68].

## Documentation

The main README in the GNRS repository (https://github.com/ojalaquellueva/gnrs/blob/ master/README.md) documents GNRS installation and configuration, command-line syntax for invoking the GNRS core service from the Linux shell, format requirements for input data, and definitions of all output returned by the GNRS. Working examples using sample data included in the repository are also provided. Example files in the API subdirectory of the GNRS GitHub repository (https://github.com/ojalaquellueva/gnrs/tree/master/api) demonstrate how to interact with the GNRS API in PHP (https://github.com/ojalaquellueva/gnrs/ blob/master/api/gnrs_api_example.php) and in R without using the GNRS R package (https:// github.com/ojalaquellueva/gnrs/blob/master/api/gnrs_api_example.R). The GNRS website includes a short tutorial on how to use the web interface (https://gnrs.biendata.org/ instructions/). Examples demonstrating usage of the GNRS R package are provided below.

## Sample workflow with the GNRS R package

### Example 1: A few political divisions

GNRS_super_simple() is the quickest method of standardizing a small number of political division names. This function does not require that the user supply a dataframe, and instead takes character vectors as input.

```
library("GNRS")
GNRS_super_simple("USA")
GNRS_super_simple(country = c("USA", "Canada"))
GNRS_super_simple(country = "USA",
  state_province = "AZ",
  county_parish = "Pima County")
```

### Example 2: Many political divisions

In most cases, users will have existing data sets containing political division names that they wish to standardize. In this case, the user only has to generate an appropriately formatted dataframe from their dataset. This can be done manually, or the function GNRS_template() can be

used to generate an empty dataframe that can then be populated. Here, we demonstrate this using the data that come packaged with the GNRS R package (accessed through the data() function).

```
data("gnrs_testfile")
gnrs_dataframe <- GNRS_template(nrow = nrow(gnrs_testfile))
gnrs_dataframe$country <- gnrs_testfile$country
gnrs_dataframe$state_province <- gnrs_testfile
$state_province
gnrs_dataframe$county_parish <- gnrs_testfile$county_parish
clean_dataframe <- GNRS
(political_division_dataframe = gnrs_dataframe)
metadata <- GNRS_metadata()
```

In both examples, the function GNRS_metadata() is usually the last step and is used to extract information that is needed for publication (e.g. version number, citation information). Complete input-output for this and the preceding example is provided in S5 Appendix.

## Case study: Validating species occurrences from the BIEN database

This example illustrates the challenges of working with political divisions from biodiversity data from multiple sources and the ability of prior name resolution by the GNRS to improve the effectiveness of downstream validations such as geopolitical validation.

### Methods

As part of the validation workflow for version 4.2 of the BIEN biodiversity observations database [16, 44] we extracted all distinct, verbatim country-, state- and county-level political divisions (declared PDCs; see "User input") from the >270 million species occurrence records in the database. The BIEN 4.2 observations consist of plant specimen records and vegetation plot data assembled from 4,946 data sources. These sources range from forest plots collected by individual researchers to complete herbarium collections databases to large aggregators of regional and global biodiversity data, with ca. 86% of observations derived from the Global Biodiversity and Information Facility (GBIF; [35]. GBIF data were obtained by downloading all occurrences in GBIF where Kingdom = 'Plantae' [69], with additional filtering to remove fossil records. Details of all data sources are provided in [45]. After extracting all declared political divisions from the BIEN occurrence records (georeferenced and non-georeferenced), we assessed the performance of the GNRS by comparing the numbers of declared political division names matching exactly to political divisions in GADM in their original form to those matching after resolution by the GNRS.

To explore the consequences of political division name resolution for downstream validation of geocoordinate accuracy, we used the subset of BIEN 4.2 species observations accompanied by geocoordinates (BIEN GSOs) to compare rates of mismatch between the declared political divisions and the political divisions indicated by the accompanying geocoordinates ("observed political divisions"). For each GSO, we determined the observed country, state and county by joining its coordinates to spatial object representations of world administrative divisions in the GADM database. Spatial joins and retrieval of GADM political division identifiers were performed using the BIEN GVS (https://github.com/ojalaquellueva/gvs; see Table 1). GADM identifiers (gid_0, gid_1 and gid_2) of the observed political divisions were then compared to the GADM identifiers of the declared political divisions. A GSO with all observed political divisions matching all corresponding declared political divisions was classified as

having passed validation; a GSO with one or more sets of non-matching observed and declared political division identifiers was classified as having failed. This validation is equivalent to testing if the GSO's coordinates fall within its declared political divisions. We performed validation twice: once using the GADM identifiers retrieved by an exact match of the verbatim declared political division name to names stored natively in the GADM database, and a second time using the GADM identifiers retrieved by resolution of declared political division names using the GNRS.

## Results & discussion

**Political division name resolution.**    A total of 409,797 unique verbatim PDCs were extracted from the 271,188,222 species observations in the BIEN 4.2 database. After processing by the GNRS, 163,174 (39.8%) of the unique PDCs were fully matched, 234,452 (57.2%) were partly matched and 12,171 (3.0%) returned no match, where "fully matched" PDCs had all declared political division names matching exactly to GADM political division names and "partly matched" PDCs had one or more unmatched political division names. Of the fully matched PDCs, only 7,593 (1.9% of total PDCs), representing 16,138,042 (6.0%) of total observations, matched exactly as submitted (that is, all verbatim political division names matched exactly to names in the GADM database). The remaining 155,581 fully matched PDCs (38.0% of total PDCs) required resolution by the GNRS—either by exact matching on codes, exact matching on alternative names and spellings, or fuzzy matching on standard and alternative names—to recover the corresponding GADM administrative units (for a breakdown of match methods at each political division level, see Tables 2–4 in S6 Appendix). Of the partly matched names, 222,987 also required some degree of resolution by the GNRS. Thus, the GNRS resolved, in part or completely, 378,568 initially non-matching PDCs (92.4% of the total PDCs), representing 232,270,686 observations, or 85.6% of the total biodiversity observation data set.

After resolution by the GNRS, 30% of country names (429), 65.6% state names (78,710) and 58.6% (175,312) of county names remained unresolved. However, unmatched country and state names represented <0.1% (263,300) and 6.2% (16,870,530) of total observations, respectively. Unmatched county-level names accounted for the majority of observations with partly or completely unresolved PDCs (72,273,914, or 26.7% of total observations). See Table 1 in S6 Appendix, for more details.

At all levels, most unmatched names appeared to be unresolvable errors such as informal region names (e.g., "Europe", "Indochina"), locality descriptions (especially in the state field), and other information unrelated to political division names (Tables 5–8 in S6 Appendix). However, many unmatched county-level names contained valid, correctly-spelled names preceded or followed by administrative level 2 type identifiers (e.g., "Oblast", "Prefecture", "District", etc.) or their abbreviations ("Obl.", "Pref.", "Distr.", "Cty.", etc.). Although the GNRS attempts to remove such class identifiers before matching, the reference tables used to detect type identifiers are incomplete; matching of county-level names could be increased by expanding these tables, in particular by adding commonly-used abbreviations.

**Geocoordinate validation with and without the GNRS.**    A total of 239,662,948 species observations had non-null values of latitude and longitude on range [-90:90] and [-180:180], respectively, allowing validation of the declared political divisions against the observed political divisions determined by their coordinates. The global distributions of points passing and failing validation, with and without prior resolution of declared political division names by the GNRS, are shown in Fig 3.

Of the GSOs passing validation, 19,123,498 (8.0% of total georeferenced GSOs) had declared political divisions correct as submitted (Fig 3B) compared to 220,762,855 (92.1%)

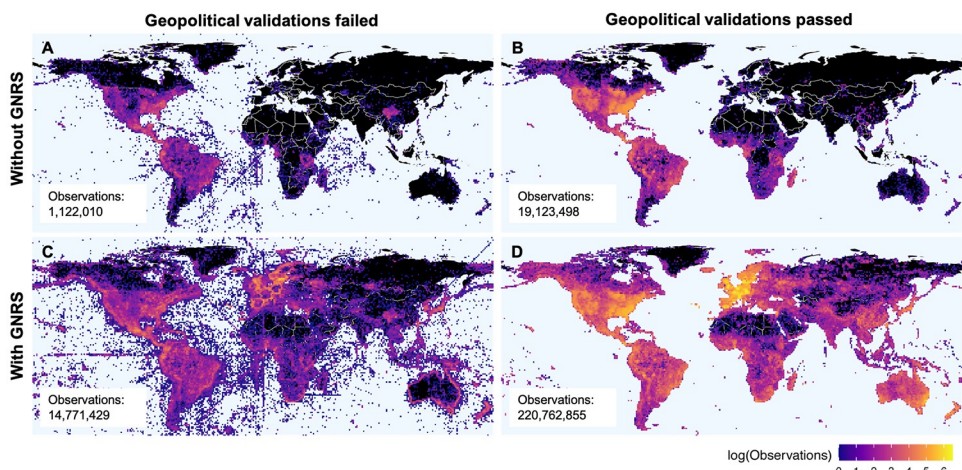

**Fig 3. Geopolitical validation results for 240 million georeferenced species occurrences, with and without prior political division name resolution by the GNRS.** Using the GNRS allows you to detect and reject an order of magnitude more bad points (A vs. C) and detect and accept an order of magnitude more valid points (B vs. D). Colors represent the density of georeferenced observations per 1 x 1 degree cells failing or passing validation that observation coordinates fall within the declared political divisions. (A) Political divisions correct as submitted, validation failed; (B) political divisions correct as submitted, validation passed; (C) political divisions correct or resolved by GNRS, validation failed; (D) political divisions correct or resolved by GNRS, validation passed. Black = zero observations.

passing validation after name resolution by the GNRS (Fig 3D). Of the GSOs failing validation, 1,122,010 (0.5%) had declared political divisions correct as submitted (Fig 3A), compared to the 14,771,429 (6.2%) erroneous GSOs detected after name resolution by the GNRS (Fig 3C). Thus, prior name resolution by the GNRS enabled the detection of an additional 201,639,357 correct observations and the exclusion of an additional 13,649,419 erroneous observations.

The increase in data validated following resolution by the GNRS had strong spatial components, with especially striking increases in Europe, Africa, Asia, and Australia. For some regions, such as western Europe and central Australia, the density of validated GSOs increased from near zero to hundreds of thousands following political division name resolution by the GNRS. Overall, name resolution by the GNRS before geopolitical validation increased the number of observations validated (passed and failed) by an order of magnitude, from 20,245,508 to 235,534,284.

## Caveats

The GNRS currently resolves to the first three levels of the political division hierarchy only (e.g., country, state, and county). For initial development of the GNRS, we chose this depth of resolution for two reasons. First, currently available biological observation data are over-whelmingly limited to three levels. For example, <2% of the plant occurrences in the GBIF download [69] used for BIEN contain declared political divisions with four levels (the maximum supported by GBIF's export schema); the remaining 98% of occurrences have three levels or fewer (S3 Appendix). Second, cross-indexing GADM and GeoName at level 4 and below would involve a disproportionate level of effort in exchange for only minuscule gains in ability to validate GSOs for species distribution modeling (our primary use case). The total number of administrative divisions at levels 4, 5, and 6 (336,907) is nearly seven times the number of divisions at levels 1–3 (49,828); level 4 alone accounts for 147,427 administrative units (S3 Appendix). Furthermore, standardized political division codes for cross-indexing GADM to GeoNames become increasingly scarce at lower lower levels, resulting in a heavy burden of ad

hoc coding and manual inspection. Limiting name resolution to the first three political division levels enabled us to potentially resolve 98% of currently available species occurrences while maintaining a realistic development timeline. However, we acknowledge that this level of resolution may be inadequate for some use cases outside of species distribution modeling. For future releases of the GNRS, we plan to extend resolution to all administrative division levels once crowdsourced initiatives to fully cross-index GADM to GeoNames [39] are substantially complete.

A second caveat to bear in mind when using the GNRS is that its data sources encompass only modern countries. For example, Yugoslavia is not present in GADM or GeoNames; historical collections from "Yugoslavia" will not be resolved by the GNRS and will not be available for downstream validation. In addition, historical changes to extant country boundaries are not represented. For example, collections from the region of South Sudan collected before 2011 (the year of South Sudan's independence from Sudan) would most likely bear the country name "Sudan". Although the GNRS will resolve the latter name, subsequent validation of the associated coordinates using GADM would locate the point of observation in modern "South Sudan", resulting in the rejection of the occurrence as invalid. We intend to release versioned updates of the GNRS database every year; upon update, changes in contemporary country boundaries and their subdivisions will be reflected automatically in GNRS name resolution results, as long as this information has been updated in GADM and GeoNames. Future development of the GNRS may address data resolution of historical administrative divisions. This challenge will require reference data that includes historical geopolitical entities and their start and end dates [70] and modern countries. Users would need to submit observation dates in addition to geocoordinates and declared political divisions.

A third caveat is that the GNRS currently resolves to GADM spatial object identifiers only. However, it also returns a variety of standard political division codes such as ISO 3166, FIPS, and HASC, which can, in turn, be used to retrieve spatial object identifiers from other widely used administrative division databases such as Natural Earth [43]. Future releases of the GNRS will store spatial object identifiers for additional sources natively within the GNRS database and expose this information to users.

## Conclusions

Political division name resolution is a critical but often neglected step in verifying the accuracy of species occurrence data. Political geovalidation is of limited value if political divisions are misspelled or represented by codes and spelling variants not present in the geospatial reference data. The GNRS fills this gap by rapidly standardizing political division names, synonyms and codes against widely-used administrative division reference data sets. A variety of interfaces enable the use of the GNRS by users with different skill levels and programming abilities—including non-programmers—and simplify integration into existing data quality pipelines. As demonstrated by a case study involving >239 million georeference species occurrences, prior name resolution by the GNRS can enable validation of an order of magnitude more error-free data and detection of an order of magnitude more erroneous data compared to using unresolved political division names and codes.

## Supporting information

**S1 Appendix. GNRS Performance features.**
(PDF)

**S2 Appendix. Building the GNRS reference database.**
(PDF)

**S3 Appendix. Administrative division hierarchy statistics for GADM and GBIF.**
(PDF)

**S4 Appendix. States-as-countries and countries-as-states.**
(PDF)

**S5 Appendix. Complete input-output for "Example: Sample workflow with the GNRS R package".**
(PDF)

**S6 Appendix. GNRS name resolution results for countries, states and counties.**
(PDF)

## Acknowledgments

We gratefully acknowledge the authors and administrators of GADM, GeoNames and Natural Earth for compiling, maintaining and distributing the data resources that made this project possible. GNRS parallelization code was originally developed by Naim Matasci for the Taxonomic Name Resolution Service [27]. The staff at NCEAS and CyVerse provided critical computational support.

## Author Contributions

**Conceptualization:** Bradley L. Boyle, Brian J. Enquist.

**Data curation:** Bradley L. Boyle.

**Formal analysis:** Bradley L. Boyle.

**Funding acquisition:** Cory Merow, Brian J. Enquist.

**Investigation:** Bradley L. Boyle.

**Methodology:** Bradley L. Boyle, Brian S. Maitner, Patrick R. Roehrdanz, Brian J. Enquist.

**Resources:** Brian J. Enquist.

**Software:** Bradley L. Boyle, Brian S. Maitner, George G. C. Barbosa, Rohith K. Sajja.

**Supervision:** Brian J. Enquist.

**Validation:** Patrick R. Roehrdanz.

**Writing – original draft:** Bradley L. Boyle, Brian S. Maitner, Erica A. Newman.

**Writing – review & editing:** Bradley L. Boyle, Brian S. Maitner, George G. C. Barbosa, Rohith K. Sajja, Xiao Feng, Cory Merow, Erica A. Newman, Daniel S. Park, Patrick R. Roehrdanz, Brian J. Enquist.

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
