## [Decision Letter · Decision Letter 0]

2 Jun 2022

PONE-D-22-11796Geographic Name Resolution Service: A tool for the standardization and indexing of world political division names, with applications to species distribution modelingPLOS ONE

Dear Dr. Boyle,

Thank you for submitting your manuscript to PLOS ONE. After careful consideration, we feel that it has merit but does not fully meet PLOS ONE’s publication criteria as it currently stands. Therefore, we invite you to submit a revised version of the manuscript that addresses the points raised during the review process.

The GNRS tool has the potential to be quite helpful for making full use of existing species occurrence data. The manuscript is generally well written and easy to follow. The overview, workflow, and examples are helpful. However, as both reviewers point out, the broader context is missing. The reviewers have provided suggestions as well as questions for consideration given challenges with naming and political divisions.

We look forward to receiving your revised manuscript.

Kind regards,

Stephanie S. Romanach, Ph.D.

Academic Editor

PLOS ONE

Journal Requirements:

2. Please amend your list of authors on the manuscript to ensure that each author is linked to an affiliation. Authors’ affiliations should reflect the institution where the work was done (if authors moved subsequently, you can also list the new affiliation stating “current affiliation:….” as necessary).

3. We note that Figure 2 in your submission contain copyrighted images. All PLOS content is published under the Creative Commons Attribution License (CC BY 4.0), which means that the manuscript, images, and Supporting Information files will be freely available online, and any third party is permitted to access, download, copy, distribute, and use these materials in any way, even commercially, with proper attribution. For more information, see our copyright guidelines: http://journals.plos.org/plosone/s/licenses-and-copyright.

a) You may seek permission from the original copyright holder of Figure 2 to publish the content specifically under the CC BY 4.0 license. 

4. We note that Figure 3 in your submission contain map/satellite images which may be copyrighted. All PLOS content is published under the Creative Commons Attribution License (CC BY 4.0), which means that the manuscript, images, and Supporting Information files will be freely available online, and any third party is permitted to access, download, copy, distribute, and use these materials in any way, even commercially, with proper attribution. For these reasons, we cannot publish previously copyrighted maps or satellite images created using proprietary data, such as Google software (Google Maps, Street View, and Earth). For more information, see our copyright guidelines: http://journals.plos.org/plosone/s/licenses-and-copyright.

a) You may seek permission from the original copyright holder of Figure 3 to publish the content specifically under the CC BY 4.0 license.  

Reviewers' comments:

Reviewer's Responses to Questions

**Comments to the Author**

1. Is the manuscript technically sound, and do the data support the conclusions?

Reviewer #1: Partly

Reviewer #2: Yes

2. Has the statistical analysis been performed appropriately and rigorously? 

Reviewer #1: N/A

Reviewer #2: Yes

3. Have the authors made all data underlying the findings in their manuscript fully available?

Reviewer #1: Yes

Reviewer #2: Yes

4. Is the manuscript presented in an intelligible fashion and written in standard English?

Reviewer #1: Yes

Reviewer #2: Yes

5. Review Comments to the Author

Reviewer #1: The manuscript describes a newly available open source data tool. The Geographic Name Resolution Service (GNRS) is fully and clearly documented in the manuscript and links are provided to existing service endpoints. The manuscript does not distinguish this service from other existing resources, and makes surprisingly little reference to a great deal of closely related previous work.

Most surprising was that the phrase "toponym resolution" does not appear in the text, and I could find no reference to any of the past several decades of work by the Natural Language Processing community on resolving geographic names.

The authors also profess no knowledge of an existing service linking databases such as GADM and Geonames, both of which are indexed by WikiData (among many other identifier systems).

Finally, the focal use case presented is for quality control of large volumes of biodiversity occurrence data. This being the case, it is surprising that no mention is made of the Global Biodiversity Information Facility, the largest available source of such data, embracing the most heterogeneous range of data sources. A sample of this data would have been the most informative option with which to test the performance of the GNRS. Also, to a reader from the biodiversity informatics community, there is an obvious analogy between the GNRS and the Global Names Architecture (GNA, https://github.com/GlobalNamesArchitecture) for taxonomic names resolution. Being used regularly on many of the same data sources presumably targeted by the GNRS, the performance of the GNA would be an interesting basis for comparison, but it is not mentioned.

I expect that the GNRS does constitute a useful new addition to the data resolution options available for biodiversity knowledge as well as other subject areas with a geographic facet. I would be more confident that it is worth trying if the authors showed more awareness of the cross-disciplinary landscape in which it will operate. The current draft of the manuscript delivers a curious impression that the work was done in isolation.

Reviewer #2: Overarching Comments

This is a really useful and important tool that is well described in this paper. The one overarching question I have is that it seems strange to me that this problem has not been encountered and potentially solved in any number of other disciplines outside of ecology/conservation such as development or policing. Can the authors provide some text discussing comparable solutions in other fields and why they were not suitable, thus requiring development of this novel tool?

Specific Comments

Line 114 It would be helpful if the authors could expand on how they maintain this neutrality given the potential political sensitivities of overlapping jurisdictional claims. As one example, I once almost completely imploded a workshop in India by presenting a map that did not show the parts of China claimed by India.

Line 132 It would be helpful to describe the full suite of BIEN tools here or in a table.

Line 193 How do countries with greater than 3 levels of political hierarchy fit in? Eg Indonesia has Country / Provinces / Kabupatians (Regencies) / Camatans (Districts) / Municipalities

Line 493 It might be helpful to add a sentence here about how the database will be updated to keep abreast of changing political jurisdictions going forward. Also, perhaps some way to automatically migrate data as the political situation changes over time?

Line 496 I appreciate the issues with historical records and would like to see the future development described here.

6. PLOS authors have the option to publish the peer review history of their article (what does this mean?). If published, this will include your full peer review and any attached files.

Reviewer #1: No

Reviewer #2: **Yes: **Nick Salafsky

---

## [Author Response · Author response to Decision Letter 0]

26 Sep 2022

Response to Reviewers

Boyle et al., "Geographic Name Resolution Service: A tool for the standardization and indexing of world political division names, with applications to species distribution modeling".

REVIEWER 1

1. The manuscript does not distinguish this service from other existing resources, and makes surprisingly little reference to a great deal of closely related previous work. Most surprising was that the phrase "toponym resolution" does not appear in the text, and I could find no reference to any of the past several decades of work by the Natural Language Processing community on resolving geographic names.

We apologize for this omission and thank the reviewer for bringing it to our attention. We have added a paragraph to the introduction in which we list key similarities and differences between the GNRS and existing services and resources. We also mention and define “toponym resolution” and highlight examples of services and algorithms developed by the NLP community and relevant to political division name resolution. Finally, we discuss why existing NLP solutions may not be optimal for the simple and somewhat different use case of toponym resolution of hierarchical structured data.

2. The authors also profess no knowledge of an existing service linking databases such as GADM and Geonames, both of which are indexed by WikiData (among many other identifier systems).

Although many services provide access to either GADM or GeoNames (for example, Geonames' own API), we have not found a service which provides comprehensive cross-indexing of both resources. Wikidata has made major progress toward indexing world administrative divisions with GeoNames identifiers, but the task is incomplete—as can be seen in the many Mexican municipalities without a value for “GeoNames ID” (e.g., Ixtlán de Juárez, https://www.wikidata.org/wiki/Q17092778). The parallel task of adding GADM identifiers has barely begun, having been proposed only in October 2020 (https://www.wikidata.org/wiki/Property_talk:P8714#Documentation). Of the 339,127 administrative divisions in GADM v3.6, only 12,371 are currently indexed in Wikidata (https://w.wiki/5bbu).

The chief obstacle to cross-indexing GeoNames and GADM is the lack of comprehensive shared identifiers linking administrative divisions at all levels. As we explain in Supporting Information Appendix 2, we were able to link the majority of administrative divisions in GADM to GeoNames using Natural Earth as a crosswalk, but many unmatched GADM entities remained at the 3rd administrative division level. Aligning these orphan administrative divisions with corresponding entities in GeoNames required extensive custom coding and manual inspection. Even relatively recent studies which attempt to combine GeoNames and GADM as sources of names and spatial objects must perform a great deal of manual cross-indexing, often with mixed results (e.g., de Rassenfosse et al., 2019, “Geocoding of worldwide patent data”. Scientific data, 6(1): 1-15.).

The key requirement for the GNRS is the availability of an integrated source of GADM identifiers, comprehensively linked to GeoNames, or at least comprehensive to the 3rd administrative division level in GADM. We have added to the introduction an explanation of why Wikidata falls short of meeting this requirement. If the reviewer is aware of another service which satisfies this comprehensive cross-index requirement, we will refactor the GNRS code to take advantage of this service and will update the manuscript accordingly.

3. Finally, the focal use case presented is for quality control of large volumes of biodiversity occurrence data. This being the case, it is surprising that no mention is made of the Global Biodiversity Information Facility, the largest available source of such data, embracing the most heterogeneous range of data sources. A sample of this data would have been the most informative option with which to test the performance of the GNRS.

We apologize for not explicitly stating that the majority of the records in the BIEN database (86% of observations) come from GBIF. The BIEN database extract used for the example contains a complete download of all GBIF records for Kingdom=Plantae, plus occurrence records from hundreds of additional data providers. These sources are enumerated in the BIEN publications cited—including the DOI of the GBIF download—but GBIF is not directly cited in our manuscript. We have added a sentence explicitly mentioning the major contribution of GBIF to the BIEN database and citing the DOI of the download used in our example analysis. We thank the reviewer for bringing this omission to our attention.

4. Also, to a reader from the biodiversity informatics community, there is an obvious analogy between the GNRS and the Global Names Architecture (GNA, https://github.com/GlobalNamesArchitecture) for taxonomic names resolution. Being used regularly on many of the same data sources presumably targeted by the GNRS, the performance of the GNA would be an interesting basis for comparison, but it is not mentioned.

This is a valid point, although we think that a closer analogy could be made between GNA and the suite of BIEN data cleaning applications, of which the GNRS is one component. This analogy was made more evident by the addition of Table 1, suggested by reviewer 2. We have added a paragraph discussing the differences and similarities between BIEN services and the GNA to section “Architecture”. 

Given their very different domains, we are less convinced of the value of a direct performance comparison between the GNRS and GNA taxonomic name resolution applications. However, the reviewer's comment alerted us to a key similarity between the GNRS and taxonomic name resolution in general: hierarchical search. This insight has helped us to clarify a subtle but important distinction between the requirements for toponym resolution in natural language text versus structured biodiversity observation data. We are grateful for this comment and the subsequent discussions it generated.

5. I expect that the GNRS does constitute a useful new addition to the data resolution options available for biodiversity knowledge as well as other subject areas with a geographic facet. I would be more confident that it is worth trying if the authors showed more awareness of the cross-disciplinary landscape in which it will operate. The current draft of the manuscript delivers a curious impression that the work was done in isolation.

We appreciate the reviewer's understandably tentative vote of confidence. We apologize for failing to mention the considerable previous research and application development on this topic. We hope that the changes discussed above and additional citations have helped remedy this failing.

REVIEWER 2

1. The one overarching question I have is that it seems strange to me that this problem has not been encountered and potentially solved in any number of other disciplines outside of ecology/conservation such as development or policing. Can the authors provide some text discussing comparable solutions in other fields and why they were not suitable, thus requiring development of this novel tool?

This is a question we asked ourselves repeatedly during development of the GNRS. Resolution of geographic place names has been the focus of research by the natural language processing community for many years. The success of this research should be apparent to anyone who uses applications such as Google Maps. However, NLP algorithms for resolving natural language text and speech are not necessarily optimal for resolving structured data such as the nested political division combinations in species occurrence data. The latter is a narrower use case whose relative simplicity allows the use of simple hierarchical searches within small, clearly defined namespaces, without having to take into account the complexities and ambiguities of natural language. This is why the GNRS correctly resolves “Mexico, Heroica Veracruz, Municipio San Andrés Tlalnelhuayocan” to a municipality, whereas Google Maps resolves it to a park.

In addition to resolving political divisions names, the GNRS returns the identifier of the spatial object needed to check for errors in the accompanying georeferenced point. Unfortunately, no existing service that we are aware of provides exhaustive cross-links between GeoNames names and GADM spatial objects. As discussed above in our response to reviewer 1's second question, Wikidata has begun linking GADM identifiers to administrative divisions but the project is far from complete.

We can only speculate as to why a political-division-name-to-spatial-object resolution service has not been developed in other research fields. One possibility, in the case of policing and crime research, at least, is that a relatively high proportion of research involves a single country only. This would result in more homogeneous datasets, diminishing the need for such a service. For the subset of researchers that are affected, however, the burden of the do-it-yourself approach is substantial (see studies cited in the manuscript). 

We have added text that we hope addresses this question. See Introduction, second to last paragraph. We thank the reviewer for raising this question and motivating us to address it more fully.

2. Line 114 It would be helpful if the authors could expand on how they maintain this neutrality given the potential political sensitivities of overlapping jurisdictional claims. As one example, I once almost completely imploded a workshop in India by presenting a map that did not show the parts of China claimed by India.

This is an important and sensitive issue, and we thank the reviewer for alerting us to deficiencies in our wording. The current claim that “the GNRS is neutral with respect to the validity of political division names and competing jurisdictional claims” could be interpreted as GNRS developers acquiescing to (and thereby tacitly endorsing) any of several controversial configurations of national and territorial boundaries currently represented in GADM and transmitted by the GNRS. We have reworded and expanded this sentence to emphasize that the GNRS has no opinion on conflicting jurisdictional claims but simply transmits as-is the representations of countries and their subdivisions provided by our data sources. We also encourage users who are dissatisfied with aspects of the current GNRS data to develop new instances of the GNRS using alternative data sources which reflect their preferences (and national legal requirements in some cases). We have moved this discussion from the Introduction to section “Reference database”, where we think it is a better fit. While we hope these changes help defuse any potential misunderstandings, we remain open to further suggestions for improvement.

3. Line 132 It would be helpful to describe the full suite of BIEN tools here or in a table.

Thank you. We have added this information as Table 1. This suggestion has helped us in several respects, including clarifying the relationship between BIEN services and the Global Names Architecture, an issue raised by reviewer 1. 

4. Line 193 How do countries with greater than 3 levels of political hierarchy fit in? Eg Indonesia has Country / Provinces / Kabupatians (Regencies) / Camatans (Districts) / Municipalities

We thank the reviewer for asking an important question which other readers will doubtless ask as well. For this initial release of the GNRS, we decided to limit resolution to the first 3 levels of the political hierarchy. This was a practical decision motivated by two main factors: (1) currently available biological observation data are overwhelmingly limited to 3 levels of political hierarchy, and (2) cross-indexing GADM to Geonames at level 4 and lower would involve a disproportionate commitment of ad hoc coding and manual inspection. We have added a new paragraph to section "Caveats" providing detailed justification for both of these claims, plus a new Supporting Information (S3 Appendix) documenting how the numbers we cite were obtained. 

In a nutshell, limiting political name resolution to the first three political division levels enabled us to potentially resolve 98% of currently available species occurrences while adhering to a realistic development timeline for the GNRS. We acknowledge that the current limit of three political division levels will be inadequate for many uses outside species distribution modeling. However, given our limited resources, we decided to postpone political division name resolution below level 3 until we can leverage Wikidata's crowd-sourced efforts to fully cross-index GeoNames and GADM.

5. Line 493 It might be helpful to add a sentence here about how the database will be updated to keep abreast of changing political jurisdictions going forward. Also, perhaps some way to automatically migrate data as the political situation changes over time?

For the GNRS we prefer to adhere to the same update schedule as the BIEN database—roughly annually. This aligns with our general philosophy of issuing infrequent major releases in the interest of reproducibility and stability of version numbers and citations. For this reason, updates triggered automatically by changes in the source data will have to remain out of scope for the GNRS. However, any changes in the source data (GADM, GeoNames, or both) will be reflected automatically in GNRS name resolution output whenever the GNRS database is updated.

We have added a sentence to this section to clarify how and when changes to contemporary country boundaries and their subdivisions will be reflected in the GNRS.

6. Line 496 I appreciate the issues with historical records and would like to see the future development described here.

Thank you. We look forward to working on this important feature request. 

EDITOR

We believe we have satisfied all PLOS ONE style and formatting requirements in the final submitted manuscript, figures and tables.

2. Please amend your list of authors on the manuscript to ensure that each author is linked to an affiliation. Authors’ affiliations should reflect the institution where the work was done (if authors moved subsequently, you can also list the new affiliation stating “current affiliation:….” as necessary).

All author names and affiliations are up to date.

3. We note that Figure 2 in your submission contain copyrighted images. All PLOS content is published under the Creative Commons Attribution License (CC BY 4.0), which means that the manuscript, images, and Supporting Information files will be freely available online, and any third party is permitted to access, download, copy, distribute, and use these materials in any way, even commercially, with proper attribution. For more information, see our copyright guidelines: http://journals.plos.org/plosone/s/licenses-and-copyright.

We require you to either (1) present written permission from the copyright holder to publish these figures specifically under the CC BY 4.0 license, or (2) remove the figures from your submission.

Regarding the copyrighted images in Figure 2—a screenshot of the GNRS web interface main page—we have replaced the globe icon with the following public domain image: https://publicdomainvectors.org/en/free-clipart/Blue-and-green-globe/54309.html. Note that the image license is CC0 1.0 (no copyright). As these Terms of Use are more permissive than CC BY 4.0 we trust that the new image will be compatible with PLOS’s license requirements. To our knowledge there was no other copyrighted material in Figure 2.

4. We note that Figure 3 in your submission contain map/satellite images which may be copyrighted. All PLOS content is published under the Creative Commons Attribution License (CC BY 4.0), which means that the manuscript, images, and Supporting Information files will be freely available online, and any third party is permitted to access, download, copy, distribute, and use these materials in any way, even commercially, with proper attribution. For these reasons, we cannot publish previously copyrighted maps or satellite images created using proprietary data, such as Google software (Google Maps, Street View, and Earth).

Regarding the potentially copyrighted map/satellite images in Figure 3, we have redone the figure using a base map from Natural Earth. Natural Earth Terms of Use are as follows:

“All versions of Natural Earth raster + vector map data found on this website are in the public domain. You may use the maps in any manner, including modifying the content and design, electronic dissemination, and offset printing. The primary authors, Tom Patterson and Nathaniel Vaughn Kelso, and all other contributors renounce all financial claim to the maps and invites you to use them for personal, educational, and commercial purposes.

No permission is needed to use Natural Earth. Crediting the authors is unnecessary.”

See: https://www.naturalearthdata.com/about/terms-of-use/.

As Natural Earth’s Terms of Use are more permissive than CC BY 4.0 (commercial use, redistribution and modification permitted; attribution not required) we trust that the new figures will be compatible with PLOS’s license requirements.

References [30]-[34] added as key citations for NLP research and application development in NLP toponym resolution. Reference [36] added in support of hierarchical searching and taxonomic name resolution. References [38]-[40] added to support claim of incomplete indexing of GADM in Wikidata. References [26] and [51]-[53] added in support of discussion of GNA relationship to GNRS. Reference [56] added; DOI citation of GBIF download used for BIEN database. Old reference [18] replaced with [20]. Old reference [29] replaced with [42]. Citations of TNRS and GNRS web user interfaces added as references [46]-[47]. Added source url to reference [66]. Numerous references have been updated to comply with PLOS format requirements for DOIs. We do not cite any retracted manuscripts.

---

## [Editor Report · Decision Letter 1]

21 Oct 2022

Geographic Name Resolution Service: A tool for the standardization and indexing of world political division names, with applications to species distribution modeling

PONE-D-22-11796R1

Dear Dr. Boyle,

We’re pleased to inform you that your manuscript has been judged scientifically suitable for publication and will be formally accepted for publication once it meets all outstanding technical requirements.

Kind regards,

Stephanie S. Romanach, Ph.D.

Academic Editor

PLOS ONE